# Next-Generation Molecular Imaging of Thyroid Cancer

**DOI:** 10.3390/cancers13133188

**Published:** 2021-06-25

**Authors:** Yuchen Jin, Beibei Liu, Muhsin H. Younis, Gang Huang, Jianjun Liu, Weibo Cai, Weijun Wei

**Affiliations:** 1Department of Nuclear Medicine, Renji Hospital, School of Medicine, Shanghai Jiao Tong University, 1630 Dongfang Rd., Shanghai 200127, China; yuchenjin@sjtu.edu.cn (Y.J.); huang2802@163.com (G.H.); nuclearj@163.com (J.L.); 2Department of Nuclear Medicine, Shanghai Sixth People’s Hospital Affiliated to Shanghai Jiao Tong University, Shanghai 200233, China; 3Human Oncology and Pathogenesis Program, Memorial Sloan-Kettering Cancer Center, New York, NY 10065, USA; 4Institute of Diagnostic and Interventional Radiology, Shanghai Sixth People’s Hospital Affiliatede to Shanghai Jiao Tong University, Shanghai 200233, China; beibei4906@163.com; 5Departments of Radiology and Medical Physics, University of Wisconsin–Madison, Madison, WI 53705-2275, USA; muhsinhy@gmail.com; 6Carbone Cancer Center, University of Wisconsin, Madison, WI 53705, USA

**Keywords:** thyroid cancer, molecular imaging, theranostics, companion diagnostics, immunoPET

## Abstract

**Simple Summary:**

Molecular imaging utilizes radionuclides or artificially modified molecules to image particular targets or pathways which are important in the pathogenesis of a certain disease. Transporter-based probes like radioiodine and [^18^F]fluoro-D-glucose ([^18^F]FDG) are widely used for diagnosing thyroid cancer (TC) and predicting the prognosis thereafter. However, newly developed probes (peptide, antibody, nanoparticle probes, and aptamer) image the fine molecular changes involved in the pathogenesis of TC and enable target-specific diagnosis and treatment of TC. Furthermore, novel molecular probes have high specificity and sensitivity, imparting a high level of objectivity to the research areas of TC.

**Abstract:**

An essential aspect of thyroid cancer (TC) management is personalized and precision medicine. Functional imaging of TC with radioiodine and [^18^F]FDG has been frequently used in disease evaluation for several decades now. Recently, advances in molecular imaging have led to the development of novel tracers based on aptamer, peptide, antibody, nanobody, antibody fragment, and nanoparticle platforms. The emerging targets—including HER2, CD54, SHP2, CD33, and more—are promising targets for clinical translation soon. The significance of these tracers may be realized by outlining the way they support the management of TC. The provided examples focus on where preclinical investigations can be translated. Furthermore, advances in the molecular imaging of TC may inspire the development of novel therapeutic or theranostic tracers. In this review, we summarize TC-targeting probes which include transporter-based and immuno-based imaging moieties. We summarize the most recent evidence in this field and outline how these emerging strategies may potentially optimize clinical practice.

## 1. Introduction

Thyroid cancer (TC) is one of the most common cancer types and its occurrence has been rapidly increasing over the last several years [1]. TC represents around 2–2.3% of new cancer cases and 0.2–0.4% of deaths from all cancer types [2,3]. In 2021, the USA may have approximately 44,280 new cases of TC and about 2200 deaths [2]. Around 90,000 new cases along with 6800 deaths were estimated in 2015 in China [2]. By 2030, TC is anticipated to become the second-most common type of cancer in females and the ninth in males [3]. 90–95% of cases present either papillary TC (PTC) or follicular TC (FTC), both of which originate from follicular cells in the thyroid and can be referred to as differentiated TC (DTC) [1].

In DTC, the thyroid retains the ability to absorb and store nearly all of the iodine in the whole body, providing the rationale for combined therapeutics with thyroidectomy and ^131^I (a beta-emitting radioisotope of iodine) therapy [4]. A total or near-total thyroidectomy removes all or most of the thyroid and DTC tissue, facilitating DTC control and the subsequent ^131^I ablation, adjuvant therapy, or therapy for DTCs [5]. The ^131^I tends to concentrate in remnant thyroid tissue, latent DTC foci, and metastatic DTC lesions. The radiation can damage the remnant thyroid tissue and DTC cells, helping DTC re-staging and improving DTC prognosis [5,6]. Unfortunately, within 10 years of an initial thyroidectomy, local recurrence and distant metastases take place in approximately 10–20% of DTCs. Notwithstanding ^131^I management, only one-third of DTCs could be regarded as having shown “complete response” with the remaining DTCs refractory to ^131^I (i.e., radioiodine refractory DTC, RR-DTC) having a poor prognosis [7].

Two less common types of TC are medullary TC (MTC) and anaplastic TC (ATC), accounting for <5% of all TC cases. Notably, however, 50–80% of MTCs show widespread metastasis at the initial diagnosis, with a five-year survival rate of 38% [8]. Furthermore, ATC is tremendously aggressive with its median overall survival of less than one year [8,9]. Thus, it is necessary to find latent lesions, precisely evaluate the grade of malignancy, adopt the most effective therapeutics, and take precautions against local recurrence or distant metastasis on time.

Traditional diagnosis methods include thyroid physical exams, blood tests (for testing biomarkers such as thyroglobulin and calcitonin), ultrasound imaging (for helping determine whether a thyroid nodule or lymph node is likely to be benign or cancerous), and other imaging tests such as CT and MRI (for TC staging and determining TC spread) [5]. In recent decades, molecular imaging (MI) has become an increasingly popular approach, applying radionuclides or artificially modified molecules to assist clinicians in locating biomarkers, potential therapeutic targets, or describing signaling pathways [10,11]. These targets play a vital role in the diagnosis and management of TC, allowing for characterization and quantification of the molecular composition of tumor tissues [12]. MI has been shown to improve diagnosis of TC, personalized management, and long-term predictive prognosis index [13]. Moreover, MI is crucial to actualizing multimodality-based theranostic strategies for TCs [14].

Over the past decade, significant progress has been made in the application of MI to TC. For instance, nanobodies and aptamers have been used to elucidate the bio-features of TC. These tracers show an antigen-binding ability resembling that of traditional antibodies [15,16,17,18]. Furthermore, immuno-single photon emission computerized tomography (immunoSPECT) and immuno-positron emission tomography (immunoPET) have encouraged the development of new theranostic methods intended for complex clinical settings, particularly for the RR-DTCs or ATCs [19]. These methods provide opportunities for obtaining deep insights into the pathogenesis of TC and as well as novel therapeutic targets for TC. Indeed, the discovery and translation of new probes enabling precise theranostics of TC are urgently needed, especially for RR-DTCs and ATCs.

Primary references are mainly derived from PubMed (available before 7 June 2021), comprehensively including the pivotal evidence in the field. In this review, transporter-based platforms are updated, and newer tracers like aptamer-, peptide-, antibody-, nanobody-, and nanoparticle-based platforms for TC are summarized. We highlight some of the potentially translatable probes in the current review. We also outline how these emerging strategies may potentially improve clinical practice.

## 2. Transporter-Targeting Probes

Most transporter-targeting probes are small-size molecules, carried into the intracellular space by transporters on the cell surface. Some transporter-associated probes may take part in cell metabolism [20,21]. Many transporter-based isotopes, including radioiodine, are routinely to image the recurrence and metastases of TC. Several alternatives to radioiodine can identify RR-DTC metastases lacking radioiodine uptake. Other transporter-based radiotracers like [^201^Tl]TlCl, [^99m^Tc]Tc-sestamibi ([^99m^Tc]Tc-MIBI), [^99^mTc]Tc-tetrofosmin, [^99m^Tc]Tc-depreotide, [^111^In]In-diethylenetriaminepentaacetic acid-octreotide ([^111^In]In-DTPA-octreotide), and [^18^F]fluoro-D-glucose ([^18^F]FDG) have been synthesized, tested, and validated as beneficial for diagnosing TC [5,22]. In particular, [^18^F]FDG has been widely applied in the management of TC [5].

### 2.1. Sodium Iodine Symporter (NIS)-Targeting Probes

NIS is the protein mainly locating at the cell plasma membrane, which carries Na^+^/I^-^ ions from the extracellular matrix into the intracellular fluid [23]. The transported iodine, as an element, helps produce thyroid hormone (iodide organification) [24]. Unlike the expression pattern of other tumor targets (low expression in normal tissues and high expression in tumor tissues), NIS is usually present at high levels in normal thyroid tissues and DTC cells, enabling radioiodine collection in normal thyroid and TC cells [25]. Nevertheless, NIS downregulation happens in RR-DTCs, poorly differentiated TCs (PDTCs), and ATCs, causing these TC cells hard to benefit much from radioiodine treatment [26].

#### 2.1.1. Radioiodine

Radioiodine, a widely used radioisotope, has a crucial role in the diagnosis and treatment of DTC. There are several medically useful radioisotopes of iodine (^125^I, ^131^I, and ^124^I, etc.). However, only ^131^I and ^124^I are commonly applied in clinical settings due to their clinically acceptable radiation half-life, diagnostic or therapeutic performance, economic cost, and safety [27]. [^131^I]NaI can track thyroid and TC cells with γ radiation on SPECT, and damage those cells by emitting β^-^ radiation [28]. [^131^I]NaI allows ablation of thyroid remnant, adjuvant therapy of TC, and therapy of TC, which vastly improves the prognosis of patients with TC [5]. ^124^I is another isotope of iodine emitting positron, which can be exploited for PET imaging. [^124^I]NaI-PET/CT has superior spatial resolution and quantification ability over [^131^I]NaI-SPECT [29].

Recently, numerous reports have focused on radioiodine for improving the diagnosis performance, and efficacy of treatment [30,31,32]. Tg tests coupled with iodine uptake assay [32], or [^124^I]NaI PET/CT only [30,31], are used for ^131^I dosimetry. Apart from performing dosimetry before ^131^I treatment, much attention should be given to increase the membranous expression of NIS, induce the concentration of ^131^I, and improve the therapeutic efficacy of ^131^I treatment. For RR-DTC, PDTC, and ATC, it is essential to explore agents that could increase NIS expression and augment the migration of NIS to the cell membrane. These agents mainly include but are not limited to, retinoic acid [33], mechanistic target of rapamycin kinase (mTOR) inhibitors [34], and very recently, V-Raf murine sarcoma viral oncogene homolog B (BRAF) and mitogen-activated protein kinase kinase (MAP2K1/2, MEK1/2) inhibitors, which inhibit the extracellular signal-regulated kinase (ERK) pathway responsible for tumor progression and radioiodine uptake [35,36] (Figure 1). RR-DTCs would be stabilized, or shrinkage after treatment with kinase inhibitors, owing to the suppressed signaling pathway and enhanced ^131^I treatment efficacy [26,37].

Lately, estrogen-related receptor gamma (ERRγ), one of the estrogen-related receptors, has gained more traction as a potential target to enhance or enable radioiodine uptake. ERRγ, a member of NR3B nuclear receptor superfamily, is a biomarker for multiple cancers, including breast cancer and prostate cancer [38]. Previous reports have shown that the ERRγ inverse agonist GSK5182 increased NIS expression and NIS-mediated iodine uptake in Kirsten rat sarcoma viral oncogene homolog (KRAS) or BRAF mutated ATC cells in vitro [39]. In addition, another ERRγ inverse agonist, DN200434, was recently shown to increase the uptake of radioiodine in ATC tumors, identifying ERRγ as a target to enhance ^131^I therapy responsiveness [40] (Figure 2). It remains to be determined if DN200434 has a re-differentiative effect in patients with either RR-DTC or ATC.

#### 2.1.2. [^18^F]Tetrafluoroborate ([^18^F]TFB)

Detecting local recurrence and metastases of DTC in radioiodine imaging is particularly important for the arrangement of local treatments, e.g., surgery or radiotherapy [5]. Whereas negative radioiodine imaging with increased serum thyroglobulin is a barrier for finding malignant lesions, the so-called “Thyroglobulin Elevated and Negative Iodine Scintigraphy” (TENIS) needs radically diverse diagnostic and therapeutic methods [26]. TENIS could be caused by poor NIS expression, iodide organification defect, or radioiodine stunning [41,42]. A failure to find NIS-expressing DTC lesions might delay the diagnosis and also the timely onset of the treatment [43]. Despite [^18^F]FDG-PET/CT could be applied for finding TENIS metastases, but its uptake might be partially caused by tumor-infiltrating immune cells [44]. Recently, [^18^F]TFB, [^18^F]Fluorosulfate ([^18^F]FS), and [^18^F]hexafluorophosphate ([^18^F]HFP) have been discovered for imaging DTCs [45,46,47,48].

[^18^F]TFB is an analog to radioiodine, having similar NIS affinities, same charge, and similar ionic radius to iodide. Therefore, [^18^F]TFB can be transported by NIS [46,49] differed from radioiodine, [^18^F]TFB can be readily synthesized at medical cyclotrons, and it provides a satisfactory half-life, dosing, biodistribution, and PET imaging quality [46,47] (Figure 3). [^18^F]TFB-PET can exclusively reveal NIS expression in tumor cells, therefore reclassifying TENIS metastases into partial or complete dedifferentiation, and helping metastasis localization and prognosis evaluation [47]. [^18^F]FS and [^18^F]HFP are two other newly discovered NIS-targeting tracers having favorable targeting efficiency, image contrasts, and biodistribution features [45,48]. Although it has been reported that FS and HFP had higher NIS affinity than TFB [45], it remains unclear if [^18^F]HFP and [^18^F]FS are superior to [^18^F]TFB. The clinical evaluation of the [^18^F]HFP and [^18^F]FS are still needed.

### 2.2. Glucose Transporter-Targeting Probes

[^18^F]FDG, mainly transported by glucose-transporter family-1 (GLUT1), is a well-known radiopharmaceutical glucose analog used in clinical PET imaging [52]. Aggressive TCs with low radioiodine uptake generally show high levels of [^18^F]FDG uptake [53]. [^18^F]FDG-PET/CT has shown great sensitivity in patients who otherwise do not benefit from ^131^I treatment. This is because metastases without radioiodine uptake tend to have high glycolytic rates, causing enhanced [^18^F]FDG uptake [54]. Furthermore, [^18^F]FDG can help detect TC recurrence or metastases and predict radioiodine uptake [54,55]. More specifically, [^18^F]FDG maximum standard unit value (SUVmax) higher than 4.0 would predict poor radioiodine uptake [54]. Besides, most PDTC, ATC, and MTC cells do not concentrate ^131^I. Thus, in these entities, [^18^F]FDG PET/CT imaging is useful in initial diagnosis, subsequent disease grading, treatment, and follow-up after treatment. Even though numerous glucose analog tracers other than [^18^F]FDG were created [56], their head-to-head comparisons with the widely used [^18^F]FDG are lacking.

### 2.3. Amino Acid Transporter-Targeting Probes

Alternatives to [^18^F]FDG are a subject of interest. Occasionally, [^18^F]FDG may yield inexplicable images. The false-positive [^18^F]FDG uptake happens in Hashimoto’s disease and Graves’ disease. In addition, it is hard to find brain metastasis on [^18^F]FDG PET/CT images because of the intense background signals [57,58]. In recent years, amino acid probes have obtained incremental attraction as alternatives to [^18^F]FDG. Amino acids probes include [^18^F]fluoro-α-methyl tyrosine ([^18^F]FAMT), [^18^F]fluoro-dihydroxyphenylalanine (^[18^F]FDOPA), L-[methyl-^11^C]-methionine ([^11^C]MET), [^18^F]fluoroethyl-tyrosine (^[18^F]FET), [^18^F]fluoroglutamine ([^18^F]FGln), and the newly discovered [^18^F]NKO-035. Of these, [^18^F]FDOPA, [^11^C]MET, and [^18^F]FGln have been investigated in TCs [59,60,61,62,63].

#### 2.3.1. [^18^F]FDOPA

[^18^F]FDOPA is a large neutral amino acid that resembles natural L-dopa, which can be transported by L-type amino acid transporter 1 (LAT1, SLC7A5) and L-type amino acid transporter 2 (LAT2, SLC7A8) [64]. [^18^F]FDOPA is a satisfactory probe for detecting MTC metastasis, persistence, and residual disease [59]. However, if [^18^F]FDOPA imaging is negative or unavailable, [^18^F]FDG should be considered, especially for aggressive MTCs displaying signs of dedifferentiation or rising carcinoma embryonic antigen (CEA) concentration in serum [59,65].

#### 2.3.2. [^11^C]MET

[^11^C]MET, transported mainly by SLC7A5, is often used to visualize parathyroid adenoma and enable focused parathyroidectomy [66,67,68]. Published data for [^11^C]MET in TC is limited. Only one case with hyperparathyroidism has been reported, which showed intense focal [^11^C]MET uptake in a cold nodule with highly increased sestamibi uptake. The nodule was finally diagnosed as FTC, indicating the incremental value of [^11^C]MET in imaging DTCs [63]. [^11^C]MET is currently being studied as a surrogate for [^18^F]FDG in other tumor types, such as brain tumors [69,70] and laryngeal cancer [71]. To date, there is no evidence showing the superiority of [^11^C]MET over [^18^F]FDG. Although complementary uptake of ^11^C-MET and [^18^F]FDG has been reported in recurrent or metastatic DTCs [72], further clarification and longitudinal study are still required to illustrate the actual value of the [^11^C]MET in clinic settings. The downside of [^11^C]MET is the short half-life of ^11^C (20.4 min), which limits its broad application [73].

#### 2.3.3. [^18^F]FGln

[^18^F]FGln, an analog of natural glutamine regulated by several glutamine (Gln) transporters (solute carrier family 1 member 5, SLC1A5; solute carrier family 38 member 1, SLC38A1; and SLC7A5; etc.), has been tested and subsequently considered as a promising probe for assessing glutamine metabolism in tumors [61]. Its use is justified by the understanding that tumor cells need extra nutrition and energy for rapid growth and proliferation, while glutamine metabolism is occasionally used by the cell as an alternative to glucose [74]. [^18^F]FGln can further complement the diagnostic capacity of [^18^F]FDG by detecting Gln metabolic changes in PTCs [62]. In [^18^F]FGln imaging, excellent contrast images can be made only 10 min after injection, while late-phase imaging (60 min) would cause a high background to some extent [62] (Figure 4).

[^18^F]FAMT, [^18^F]FET, and the newly reported [^18^F]NKO-035 are all transported by L-type amino acid transporters, which are overexpressed in tumor cells [21,75]. However, data for those probes remain inadequate now. Furthermore, unlike other amino acid tracers transported by multiple unspecific amino acid transporters, [^18^F]FAMT has an α-methyl moiety that allows it to be exclusively specific to SLC7A5, making it highly tumor-specific [76,77]. Furthermore, [^18^F]FAMT is more specific for tumors than [^18^F]FDG, although their sensitivities are similar. However, [^18^F]FAMT imaging is comparable to [^18^F]FDG imaging in diagnosing tumors other than TCs [20]. Future studies are warranted to investigate the amino acid metabolism in TCs and the diagnostic value of amino acid tracers in large cohorts.

### 2.4. Nucleoside Transporter-Targeting Probes

Radiolabeled or fluorescent nucleobase analogs are currently used to diagnose solid tumors, including cancers of the bladder, breast, lung, ovary, and pancreas. Regarding diagnosis of TC specifically, only [^18^F]fluorothymidine ([^18^F]FLT) has been tested to date. [^18^F]FLT, which can be taken up by equilibrative nucleoside transporter 1 (ENT1), is a marker of cell proliferation [78]. In one study, 20 DTCs were assessed with [^18^F]FLT and [^18^F]FDG on PET/CT. While 69% of the metastatic lesions were identified by focal increases in [^18^F]-FLT uptake, a lower result than the 92% identified by [^18^F]FDG PET/CT. It is also demonstrated that [^18^F]FDG has the advantage in terms of specificity and accuracy over [^18^F]FLT in finding local lymph node malignancy and distant metastases [79]. So far, [^18^F]FLT PET/CT has not progressed very far in diagnosing TCs.

## 3. Peptide-Based Probes

Peptide tracers have played vital roles in MI due to their unique advantages, notably their low molecular weight and ability to bind tumor biomarkers specifically, with low toxicity to surrounding non-cancer cells. Multiple tracers, like [^68^Ga]Ga-dodecane tetraacetic acid labeled RGD2 ([^68^Ga]Ga-DOTA-RGD2; RGD: Arg-Gly-Asp), [^68^Ga]Ga-prostate specific membrane antigen ligand ([^68^Ga]Ga-PSMA) with conjugates of N,N’-bis[2 -hydroxy-5-(carboxyethyl)benzyl]ethylene diamine-N,N′-diacetic acid (HBED-CC) or DOTA, [^68^Ga]Ga-DOTA-DGlu-Ala-Tyr-Gly-Trp-(N-Me)Nle-Asp-1-Nal-NH2 ([^68^Ga]Ga-DOTA-MGS5), [^111^In]In-DTPA-octreotide, and other somatostatin analogs have been developed for imaging TCs, particularly MTCs and RR-DTCs [80].

### 3.1. Somatostatin Receptor (SSTR)-Targeting Probes

Somatostatin receptors have become typical therapeutic targets in neuroendocrine tumors (NETs) because they are often overexpressed on the surface of tumor cells. This has led to the development of several ^68^Ga-labelled somatostatin analogs as PET imaging probes [81], which could be used for the diagnosis of MTC [80]. ^68^Ga-labeled somatostatin analogs, including [^68^Ga]Ga-DOTA-(1-Nal^3^)-octreotide ([^68^Ga]Ga-DOTANOC), [^68^Ga]Ga-DOTA(0)-Phe(1)-Tyr(3)-octreotide ([^68^Ga]Ga-DOTATOC), and [^68^Ga]Ga-DOTA-(Tyr3)-octreotate ([^68^Ga]Ga-DOTATATE), are valuable diagnostic tools showing excellent performance in the majority of patients with NETs [82]. Nevertheless, studies reporting the diagnostic value of SSTR-targeted PET in recurrent MTC are limited. A meta-analysis involving nine studies reported that the tumor detection rate on SSTR-based PET or PET/CT is only 63.5% in recurrent MTC, which is lower than that in other NETs [83].

### 3.2. αvβ3 Integrin-Targeting Probes

The integrin αvβ3 expression on epithelial cells and mature endothelial cells is relatively low, however, it is commonly and highly expressed in solid tumors. RGD and RGD_2_ are peptides that bind integrin αvβ3 [84]. Recently, the dimeric [^68^Ga]Ga-DOTA-RGD_2_ has been successfully applied for PET imaging of RR-DTCs in clinical settings [85], showing sensitivity, specificity, and accuracy of 82.3%, 100%, and 86.4%, respectively, which exceeds the same measurements in [^18^F]FDG of 82.3%, 50%, and 75%, respectively. For RR-DTCs, the advantage provided by [^68^Ga]Ga-DOTA-RGD2 is the ability to detect lesions not detected by [^18^F]FDG [85] (Figure 5). Furthermore, diagnosis of RR-DTCs using [^68^Ga]Ga-DOTA-RGD_2_ is better accompanied by [^177^Lu]Lu-DOTA-RGD_2_, a potential treatment option for RR-DTCs [86]. Considering that [^68^Ga]Ga-DOTA-RGD_2_ and [^177^Lu]Lu-DOTA-RGD_2_ are a useful theranostic pair for RR-DTCs, the potential to improve the theranostic landscape of RR-DTCs by sequentially using these agents is high. Nuclear medicine approaches have revolutionized the theranostic arsenal for DTCs, and we are confident that there is room to optimize the management of RR-DTCs with these novel agents.

### 3.3. PSMA-Targeting Probes

PSMA is overexpressed on the prostate cancer cell membrane. Recently, several studies found unexpected PSMA-targeted radiotracer uptake by TCs, including RR-DTCs [87,88,89,90,91] (Figure 6). In addition, ~50% of TC microvessels showed high expression of PSMA related to tumor size and vascular invasion [89]. Thus, it is reasonable that high-grade TCs can be targeted by PSMA-specific radioligands like [^177^Lu]Lu-PSMA and [^225^Ac]Ac-PSMA [92], establishing a novel theranostic platform for TCs that are refractory to radioiodine treatment. Currently, the clinical interest and focus of PSMA-targeted theranostics remain primarily oriented towards prostate cancers. It is worth exploring the performance of PSMA-targeted agents in RR-DTCs. The authors wonder if PSMA-targeted agents will open a new horizon for RR-DTCs in the future.

### 3.4. Cholecystokinin-2 Receptor (CCK2R)-Targeting Probes

CCK2R is highly expressed in 90% of MTC, 50% of small cell lung cancers, 60% of astrocytomas, insulinomas, stromal ovarian cancers, gastrointestinal stromal tumors, and more than 20% of gastroenteropancreatic tumors [93,94,95]. As a new peptide tracer targeting CCK2R, ^68^Ga-DOTA-MGS5 is supposed to be superior to [^18^F]FDOPA in diagnosing MTCs. The comparison between the [^68^Ga]Ga-DOTA-MGS5 and [^18^F]FDOPA was performed in a 75-year-old female patient with recurrent MTC (calcitonin: 2726 ng/l), who underwent consecutive [^68^Ga]Ga-DOTA-MGS5 and [^18^F]FDOPA-PET/CT. [^68^Ga]Ga-DOTA-MGS5 found three obvious liver lesions with higher uptake than [^18^F]FDOPA-PET/CT (SUVmax = 6.4–8.3 vs. 3.7) and showed a good lesion-to-background contrast in the liver, which might yield complementary information to [^18^F]FDOPA-PET in patients with recurrent MTC [96] (Figure 7). The background of [^68^Ga]Ga-DOTA-MGS5 seems higher than that of [^18^F]FDOPA in our opinion.

## 4. Antibody-Based Probes

Antibodies are high-affinity molecules with strict targeting abilities that are used for highly specific binding [97]. The development and translational use of antibody therapeutics have shaped the model of molecular targeted therapy and immunotherapy. The high affinity of monoclonal antibodies for their targets promotes the rational and efficacious use of antibody therapeutics [98]. We have advocated that PET imaging with radiolabeled antibodies or antibody fragments (i.e., immunoPET) provides a powerful platform for visualizing the tumor targets, selecting suitable patients for targeted therapies or immunotherapies, and assessing the therapeutic responses thereafter [19]. The first-generation monoclonal antibodies (mAbs) were of murine origin, making them immunogenic, limited for their clinical use. Consequently, chimeric mAbs, humanized mAbs, and complete human mAbs were produced to solve this issue [98]. One limitation of the full-size antibody probes is their considerable size (~150 kDa), which leads to a long circulatory half-life and reduced tissue penetration [99]. To ameliorate the imaging quality and efficiency and accelerate clinical translation, some smaller molecule substitute probes have been investigated, including antigen-binding fragments (Fabs) and engineered Fab variants, single-chain variable fragments (scFv), diabodies, minibodies (~25–100 kDa), and other types of therapeutic proteins, such as affibodies and nanobodies [19]. Facilitated by these developments, multiple antibodies, and antibody derivatives have been designed as either imaging probes or therapeutic agents to induce cancer cell death and elicit host immune effector responses in TC [19].

### 4.1. Single Target Immunoglobulin G (IgG) Probes

Full-size IgG antibody probes have been applied to tumor detection, staging, guidance of local treatment, identification or validation of tumor targets, and assessment of therapeutic response or tumor prognosis [100]. Once the first-rank antigen has been selected, the corresponding IgG can be labeled with a radionuclide or fluorescent tag [19]. The radionuclide labeled IgG can be visualized via immunoPET imaging, and the fluorescent tags can be visualized through the fluorescence system during thyroidectomy or metastasectomy [101].

#### 4.1.1. Epidermal Growth Factor Receptor (HER2, ERBB2)-Targeting Probes

The human HER2, which is expressed on the cell plasma membrane [102], is a typical molecular marker for breast cancers and a subset of aggressive thyroid cancers [103,104]. HER2 overexpression was found in 44% of FTCs, 18% of PTCs [105], and certain ATCs [106]. Several HER2-specific agents—such as trastuzumab, lapatinib, and pertuzumab—have primarily ameliorated the prognosis in HER2-positive breast cancers [107,108]. Outside of breast cancers and TCs, HER2 is also widely overexpressed in multiple malignancies, including bladder, pancreatic, ovarian, and stomach cancers [109,110,111]. CUDC-101 (an inhibitor of epidermal growth factor receptor (EGFR), HER2, and histone deacetylase (HDAC)) inhibited tumor growth and metastases in metastatic ATC models [112], and lapatinib (an inhibitor of HER2 and EGFR) overcame the ERK and v-akt murine thymoma viral oncogene homolog 1 (AKT) rebound in PLX4032 resistant TC cells [113]. These studies indicate that HER2 is a potential target for developing theranostic interventions for advanced TCs.

By labeling the HER2-targeting mAb pertuzumab with ^89^Zr, we have developed the [^89^Zr]Zr-DFO-pertuzumab and evaluated its diagnostic efficacy in subcutaneous and orthotopic ATC models [114] (Figure 8). ImmunoPET and fluorescence imaging indicated that radiolabeled or fluorescence-labeled HER2 probes are promising for the management of ATCs, which may become helpful tools for image-guided tumor removal or identifying HER2-positive ATCs for HER2-targeted therapies. However, clinical studies are needed for further translation. To facilitate clinical translation and broad clinical use, we have developed a series of novel nanobody-based tracers to delineate HER2 expression. We will test the performance of the tracers in TC models very soon.

#### 4.1.2. Intercellular Adhesion Molecule-1 (ICAM-1, CD54)-Targeting Probes

ICAM-1, belonging to the immunoglobulin superfamily of cell adhesion molecules, consists of five extracellular IgG-like domains and one cytoplasmic tail [115]. ICAM-1 is found to be expressed at low levels in normal tissue, but at high levels in multiple types of cancer, including TCs [116,117]. One of its important features is that it can initiate tumor transmigration and invasion [116,118]. Furthermore, ICAM-1-targeted chimeric antigen receptor T (CAR-T) cells can robustly kill TC cells [119,120]. Research thus far has suggested ICAM-1 as an ideal target for TC diagnosis and treatments. For this purpose, Wei et al. created an immunoPET probe [^64^Cu]Cu-NOTA-ICAM-1, which targets ICAM-1. [^64^Cu]Cu-NOTA-ICAM-1 immunoPET imaging showed high contrast in diagnosing the subcutaneous and orthotopic ATCs in preclinical settings [101] (Figure 9). With the published data and unpublished data in hand, we believe that ICAM-1 may serve as a viable biomarker for certain types of TCs. However, it remains to see the diagnostic utility of ICAM-1–targeted tracers in patients with TCs.

#### 4.1.3. Lectin Galactoside-Binding Soluble 3 (LGALS3, Galectin-3, or Gal3)-Targeting Probes

Gal-3 is a protein that is undetectable in normal and benign thyroid tissues but highly expressed in DTC cytosol, cell membranes, and intercellular substance [121]. The expression of galectin-3 as a biomarker for TCs has been validated in two multicenter studies [122,123]. The sensitivity and specificity of Gal-3 immunodetection reached 94% and 98% in distinguishing benign from TC lesions, with positive and negative predictive values of 98% and 94%, respectively, and diagnostic accuracy of 96% [122]. [^89^Zr]Zr-labeled Gal3 mAb ([^89^Zr]Zr-DFO-Gal3) or Gal-3 mAb with F(ab’)_2_ conjunction ([^89^Zr]Zr-Gal3-F(ab’)_2_) has shown good binding to TC in vivo, allowing it to be potentially used for the detection of recurrence and metastases [124,125] (Figure 10). The particular design of [^89^Zr]Zr-DFO-Gal3-F(ab’)_2_, a protein formed of two F(ab’) fragments, results in faster blood clearance and lower liver uptake than traditional mAb-based tracers [125]. The high uptake of [^89^Zr]Zr-DFO-Gal3-F(ab’)_2_ in kidneys is due to the urinary excretion [125], which should not be problematic because metastatic TC to the kidney is very rare [126]. For a diagnostic purpose, the dose is usually quite low and so is the nephrotoxicity. Thus [^89^Zr]Zr-DFO-Gal3-F(ab’)_2_ might be an excellent candidate for translation into the preoperative evaluation and postoperative follow-up.

### 4.2. Bispecific IgG Probes

Bispecific antibody (BsAb) probes have filled the vacancy of single target IgG probes in theranostics by providing higher antigen-binding capacity in tumor tissues than the monomeric counterparts [19]. Additionally, the pharmacokinetics of BsAbs could be improved by protein modification. The ability of BsAbs to bind to two targets allows these bispecific IgG probes to display an enhanced role for targeting two antigens on a tumor cell surface, linking the tumor cells and immune cells, for instance [127,128]. However, until recently, only one BsAb targeting CEA and HSG has been thoroughly investigated in the diagnosis of MTC.

As stated previously, the intense expression of CEA is a biomarker of MTC. Prior clinical studies have shown the high sensitivity of the combination of anti-CEA BsAbs and ^111^In or ^131^I labeled haptens-peptides [129,130,131]. IMP288, an HSG hapten, was reported to have the ability to bind multiple radionuclides [132]. Meanwhile, a trivalent BsAb (called TF2), was engineered composing one HSG glycine Fab fragment and two anti-CEA Fab fragments [133]. The combination of ^68^Ga labeled IMP288 and TF2 in PET imaging yields high sensitivity and specificity; Nevertheless, the pretargeting conditions may still need to be modified to reduce or avoid IMP288-induced adverse effects (malaise, bronchospasm, tachycardia, and hypertension) [132,134]. The delivery method of the tracer may challenge patients’ acceptability because the combination of IMP288 and TF2 requires two injections: the first injection for TF2 BsAb, and a second injection for [^68^Ga]Ga-IMP288, with a time lag (one or two days) between the two injections [134] (Figure 11). The pretargeting strategy was used for the diagnostic purpose in the study. Replacement of ^68^Ga with beta-emitter (e.g., ^177^Lu) or alpha-emitter (e.g., ^225^Ac) will further develop pretargeting therapeutic strategies, which will hopefully maximize the therapeutic index and minimize the adverse effects.

### 4.3. Fab-Based Probes

Fab is characterized by a light chain and a heavy chain of an immunoglobulin, containing variable regions, constant domain of the light chain (CL), and first constant domain of the heavy chain (CH1) [135]. The Fab, therefore, takes the specificity of the immunoglobulin. Unlike the traditional antibodies (produced from mammalian cells), Fabs could be generally and easily produced from bacteria cells, like *E. coli* [136]. One drawback of Fab is the limited retention on the antigen and rapid clearance [137]. Some Fabs have been discovered for the treatment of TC (targeting cluster of differentiation 276 [CD276] [138], etc.), and some publications reported the potential value of Fab as diagnostic probes targeting Galectin-3 [125,139,140].

[^89^Zr]Zr-DFO-αGal3-Fab-PAS_200_, an imunoPET probe fused with 200 Pro, Ala, and Ser residues (PAS_200_) and conjugated with [^89^Zr]Zr-deferoxamine ([^89^Zr]Zr-DFO), is a recently reported Fab-based probe derived from the rat anti-Gal3 mAb. Similar to the full-size [^89^Zr]Zr-labeled Gal-3 mAb (mentioned in Section 4.1.3) [125], the [^89^Zr]Zr-DFO-αGal3-Fab-PAS_200_ can bind the Gal-3 well [139] (Figure 12). Unlike the uptake of full-size ^89^Zr-labeled Gal-3 mAb which lasts over five days after injection, the [^89^Zr]Zr-DFO-αGal3-Fab-PAS_200_ was supposed to have a shorter lasting time, but the exact time was undetermined [125]. Research concerning a head-to-head comparison between the anti-Gal3 IgG probe and the corresponding fragment probe is lacking.

### 4.4. Nanobody-Based Probes

A single-domain antibody (sdAb, nanobody) is an engineered antibody fragment containing a single monomeric variable antibody domain. Compared to the large size of full-size antibodies (~150 kDa), nanobodies (~15 kDa) can be delivered to tumors with comparatively less obstruction [141]. Nanobodies can be reconstructed to Fc-domains or conjugated to molecular inhibitors, radioisotopes, fluorescent dye, and nanoparticles, making them suitable for targeting tumors with many applications [142]. For example, Jailkhani et al. established nanobody libraries against extracellular matrix (ECM) proteins, which are hallmarks of many diseases, including cancers. PET/CT imaging showed that ^64^Cu-labeled NJB2 nanobody probes targeted ECM and detected breast cancer and melanoma *for* primary and metastatic foci (including thyroid) with excellent contrast [143]. Thus, nanobody probes may open up a promising opportunity for application in TCs. So far, nanobody probes remain absent in TC research [144]. Our team has developed a series of nanobodies targeting various targets (e.g., tumor-associated calcium signal transducer 2 [TACSTD2, TROP-2], ICAM-1, integrin associated protein [CD47], and melanoma cell adhesion molecule [MCAM, CD146]) and are fully exploring the theranostic potential of the nanobodies in TCs.

## 5. Other Probes

### 5.1. Aptamer-Based Probes

Aptamers are nucleic acids with antigen selectivity rivaling that of antibodies [145]. They bind to their target through electrostatic interactions, hydrophobic interactions, and induced fitting. Aptamers also offer target recognition that is comparable to traditional antibodies. Unlike antibodies, however, aptamers can be produced more feasibly. Its additional advantages include favorable storage properties and limited immunogenicity in vivo [146]. The major drawback of aptamers is the lack of stability in vivo. Regarding their application in TC, only a few aptamer probes have been reported [15].

#### 5.1.1. Prominin 1 (PROM1, CD133)-Targeting Probes

CD133 is a kind of glycoprotein mainly expressed in hematopoietic stem and progenitor cells [147]. As a marker of cancer stem cells of brain tumor, colon cancer, melanoma, and ATCs [148,149,150,151], it is known to be responsible for the rapid growth of ATC and PTC cells [152,153]. Ge et al. synthesized and characterized an aptamer AP-1-M targeting CD133 in an ATC xenograft model. The synthesized AP-1-M-doxorubicin conjugates can effectively bind CD133-expressing tumor cells, and an intense signal may reflect the tumor proliferation at a fast pace [15] (Figure 13).

#### 5.1.2. PTC Tissue-Targeting Probes

Zhong et al. generated a PTC tissue-specific aptamer (TC-6) via tissue-based systematic evolution of ligands by exponential enrichment (SELEX), with clinical PTC tissues (positive control) and non-tumor thyroid tissues (negative control). The TC-6 can specifically distinguish PTC from other non-tumor tissues (Figure 14), and suppress the migration and invasion of PTC cells [16]. However, the exact molecular target remains unknown.

### 5.2. Nanoparticles-Based Probes

Nanoparticles have been emerging with widespread attention in MI, drug delivery, and disease treatment. Nanoparticles have brought their potential as MI agents to TC, primarily through their applicability in fluorescence imaging, ultrasound, and MRI [154,155]. These modalities enable nanoparticles to accumulate in cells by activation through US, light, temperature, and pH change, depending on the nanoparticle structures and their surface molecules. The ligand options for targeted nanoparticles are somewhat limited. Antibodies and peptides are the primary ligand choices due to their specific affinity to targets in TC fields. Although applications of targeted nanoparticles in TC have so far been limited, there have been publications investigating nanoparticles conjugated to antibodies targeting epidermal growth factor receptor (EGFR) or Src homology 2 (SH2) domain-containing phosphatase 2 (SHP2) [154,155].

#### 5.2.1. EGFR-Targeting Probes

EGFR is a receptor binding the extracellular epidermal growth factor family (EGF family) [156]. In many tumor types, including TC, increased EGFR expression or activity initiates the tumor cell progression [112]. Recently, EGFR has been the target of the newly created nanoparticle (called C-HPNs) based on a core-shell system loaded with EGFR-targeted cetuximab and 10-hydroxycamptothecin (10-HCPT, chemotherapy agent). The EGFR antibody ligands enable nanoparticles to attach to cells which overexpress EGFR. With low-intensity focused ultrasound (LIFU) assistance, the liquid perfluoropentane (PFP) core in the nanoparticles would become vaporized and transformed into microbubbles, enhancing ultrasound contrast for tumor diagnosing. The core explosion induced by PFP boiling causes the release of 10-HCPT, providing more targeted delivery of the chemotherapeutic drug [154] (Figure 15).

#### 5.2.2. Protein Tyrosine Phosphatase Non-Receptor Type 11 (PTPN11, SHP2)-Targeting Probes

Another example is the SHP2, which is a tumor biomarker, acting as a signal of cell proliferation and immortality [157]. Hu et al. created an SHP2-targeted core-shell nanoparticle chelated with the contrast agent Gd^3+^ on the surface (NPs-SHP2). Similar to the EGFR-targeted nanoparticles mentioned previously, PFP-based LIFU can facilitate the probe carrying contrast agent been accumulated in the thyroid tumor area for enhanced MRI [155] (Figure 16).

## 6. Conclusions and Future Perspectives

In summary, MI plays a vital role in evaluating and managing TCs, especially in accurately finding occult foci that are undetected by traditional ultrasound, CT, and MRI, thereby helping TC patients get the precise therapeutics (Table 1). Transporter-based probes tend to have high sensitivity, and immune-based probes generally have high specificity. So far, there is no single probe to reveal all the lesions both specifically and sensitively. Moreover, although TCs can be classified in a single pathological category, the biomarker expression in TC can vary dramatically. The apparent heterogeneity of TC requires the availability of more than one therapeutic method. By fully elucidating the biological characteristics of TCs and thoroughly exploring the biomarkers enriched in TCs [26,158,159], we believe that we can discover helpful targets for developing diagnostic probes and companion therapeutic agents for different molecular types of TC, not limited to pathological typing and phenotyping.

With the development of biological techniques and imaging tools, more valuable imaging methods are emerging. Integration of multiple imaging modalities and anatomical features would help physicians diagnose and treat TCs in a timely target-specific manner. Nanoparticles may enable anti-TC drug delivery and multimodality imaging, which may further improve the management of TC. However, the authors are cautious because of the limited clinical evidence in the field of TC. Bispecific MI probes have gained more traction in the past decade. The synthesis of bispecific antibody or antibody fragment tracers has been thoroughly elucidated elsewhere [170]. The importance of bispecific probes is that they enable enhanced affinity and high image quality, providing the ability targeting more occult foci than traditional single-target probes, and inducing more comprehensive application across TC patients with complex biological characteristics.

For antibody probe design, the traditional antibody-based radioactive or fluorescent probes for TCs have simply been prepared by non-selective conjugation on lysine/cysteine residues with or without chelators [171,172]. This approach may reduce target binding affinity, especially at a high conjugate/protein ratio, and in any case, leads to a mixture of products with different numbers of tags per protein molecule [173]. The method may also cause undesirable biodistribution (e.g., high kidney uptake and poor tumor targeting due to in vivo cleavage of the S–S linkage) and pharmacokinetics (fast antibody clearance for modification of interchain disulfide cysteine) [174]. Protein engineering techniques are progressing very rapidly. In this setting, the site-specific and homogeneous introduction of the tags into the targeting moieties would be more advantageous. The emerging techniques mainly include chelator conjunction to the antibody glycan region, enzyme-assisted chelator attachment, and incorporating and chelating radioisotope into amino acid sequences [171]. This may help us design molecular imaging agents and companion therapeutic agents with increased possibility for clinical translation.

For the diagnosis of TC, the superiority of MI over conventional anatomical imaging is clear, with advantages of favorable spatial and temporal resolution and functional imaging [175]. The emergence of MI has fundamentally changed the management of TC. For instance, [^18^F]FDG PET/CT plays its role in optimizing initial therapy, which is mandatory for improving DTC outcome. [^18^F]FDG PET/CT could be conducted if some foci are radioiodine non-avid before treatment planning. Rosenbaum-Krumme et al. found that the TNM staging and management were changed from standard therapy (surgery plus ^131^I therapy) to individual therapy (standard therapy plus external beam therapy or targeted therapy etc.) because of the [^18^F]FDG PET/CT results in 21% of the high-risk DTCs [176]. We suppose that MI in TCs would become a helpful modality for tumor staging, prognosis evaluation, which may lead to immense changes in what treatments the TC patients are given, maximizing the benefits of individual therapy.

For TC therapy, despite the rapid progress in molecular imaging, we firmly believe that the innovation of therapeutic agents should accompany the development of diagnostic agents. For example, radionuclides such as ^177^Lu, ^225^Ac ^188^Re, ^67^Cu, ^47^Sc, ^166^Ho, ^90^Y, ^161^Tb, ^149^Tb, ^212^Pb, and ^213^Bi emitting α-particles, β-particles, and Auger electrons can be feasibly chelated with DOTA, NOTA, etc., which is similar to the diagnostic isotopes mentioned in this review, ^64^Cu or ^68^Ga chelated with NOTA or DOTA [177]. For theranostic application, diagnostic probes with a chelating agent and a radionuclide suitable for imaging (e.g., NOTA and ^64^Cu) can be used to map the target and assess the therapeutic potential of the same probe labelled with the same chelating agent and a therapeutic radionuclide (e.g., NOTA and ^67^Cu) [19]. ^149^Tb is another theranostic radioisotope that has not been investigated in thyroid cancer research to date. ^149^Tb can simultaneously emit positrons (β^+^ particles), α-particles, and γ-radiation, allowing PET and α particle-based therapy to go on at the same time [177]. The current review focuses on emerging probes for imaging with examples. Further introduction and discussion about the concept of TC therapy or theranostics will be updated and illustrated in an upcoming review. We hope the ever-developing diagnostic and therapeutic probes and theranostic applications can substantially improve the management of TCs, especially the aggressive RR-DTCs, MTCs, and ATCs.

In the last two decades, there has been a general increase in the prevalence of TC. The phenomenon is partially due to environmental factors (e.g., chemical pollution, anthropogenic or natural radiation), but is also due to overdiagnosis by increased screening with more sensitive methods (e.g., high-resolution ultrasound), especially in developed countries (e.g., South Korea and the United States), leading to unnecessary treatment. As mentioned previously, most incident TCs are low-risk DTCs that tend to retain their stability over the years until death due to aging. Therefore, it is unnecessary to treat indolent TCs because of the low cost-effectiveness and the practically unchanged mortality. Something to note is that the ever-developing MI techniques might more sensitively detect TCs, potentially causing overdiagnosis for indolent TC and overtreatment via invasive methods (e.g., thyroidectomy or metastasectomy) or noninvasive therapeutics (e.g., chemotherapy agents or multikinase inhibitors). To avoid these pitfalls, future work could focus on discovering prognosis- or progress-related targets or probes, thereby classifying TCs into indolent and active disease. In other words, future MI research should not be limited to the field of finding latent TC lesions.

## Figures and Tables

**Figure 1 cancers-13-03188-f001:**
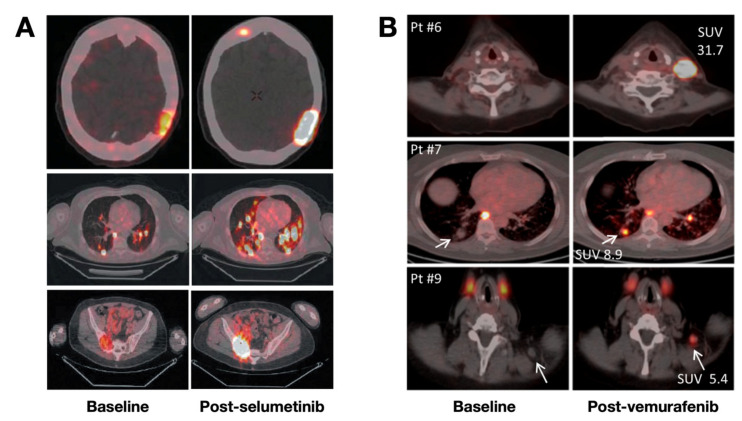
[^124^I]NaI PET/CT images of patients with RR-DTC or PDTC with or without kinase inhibitors. (**A**) PET/CT images showed enhanced iodine uptake of lesions post-treatment with selumetinib in nearly all previously negative head, lung, and sacroiliac bone metastases. Reproduced with permission from [35], copyright 2013 Massachusetts Medical Society. (**B**) PET/CT images showed enhanced radioiodine uptake of lesions in the neck and lung after treatment with vemurafenib, a specific BRAF^V600E^ inhibitor. Reproduced with permission from [36], copyright 2019 Endocrine Society.

**Figure 2 cancers-13-03188-f002:**
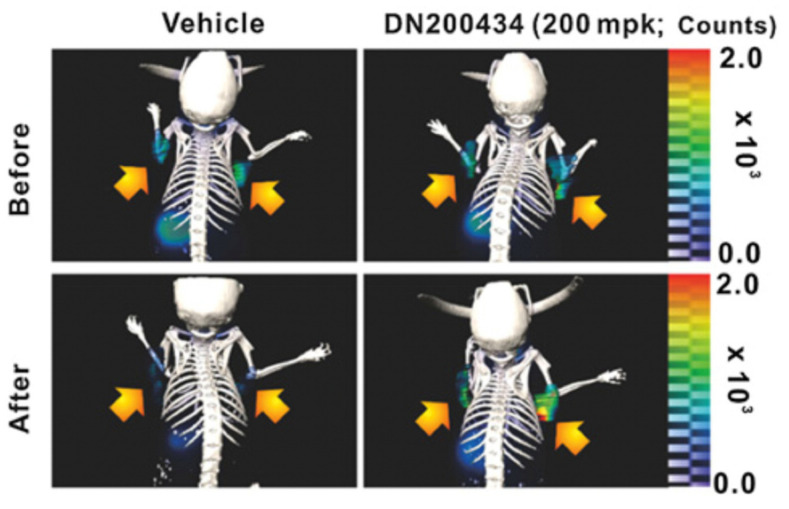
[^124^I]NaI-PET/CT demonstrates enhanced iodine uptake in CAL62 ATC tumor after treatment with DN200434. The arrows indicate the ATC tumor. Reproduced with permission from [40], copyright 2019 American Association for Cancer Research.

**Figure 3 cancers-13-03188-f003:**
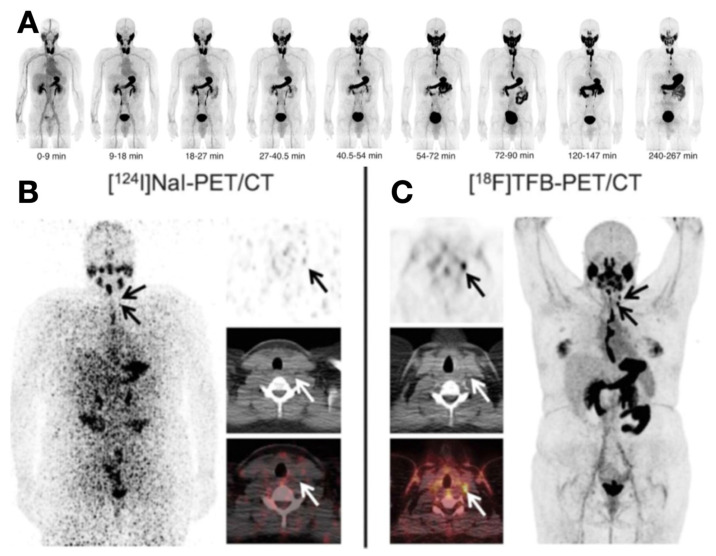
Evaluation of thyroid cancer via [^18^F]TFB-PET/CT and [^124^I]NaI-PET/CT. (**A**) Biodistribution of [^18^F]TFB at various time points on PET images. [^18^F]TFB accumulated rapidly in the thyroid and other normal tissues like the salivary gland and stomach within 10–30 min. Reproduced with permission from [50], copyright 2017 Society of Nuclear Medicine and Molecular Imaging. (**B**,**C**) A comparison between [^124^I]NaI-PET/CT and [^18^F]TFB-PET/CT in a 26-year-old patient post-thyroidectomy. (**B**) [^124^I]NaI-PET/CT was unremarkable for PTC. (**C**) In contrast, [^18^F]TFB-PET/CT revealed two foci in the left lateral cervical region. Reproduced with permission from [51], copyright 2018 Wolters Kluwer Health.

**Figure 4 cancers-13-03188-f004:**
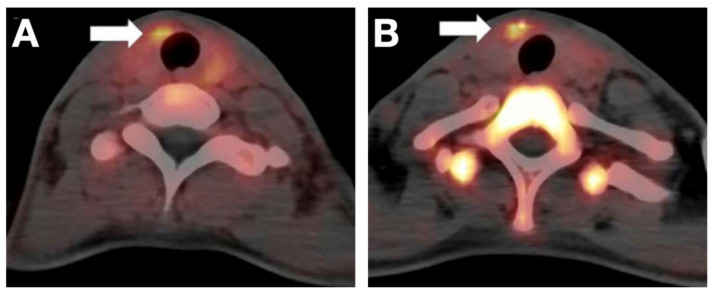
Example of a PTC revealed by [^18^F]FGln at 10 min (**A**) and 60 min (**B**) post-injection, respectively. The arrows indicate the malignant lesion. Reproduced with permission from [62], copyright 2020 Springer Nature Inc.

**Figure 5 cancers-13-03188-f005:**
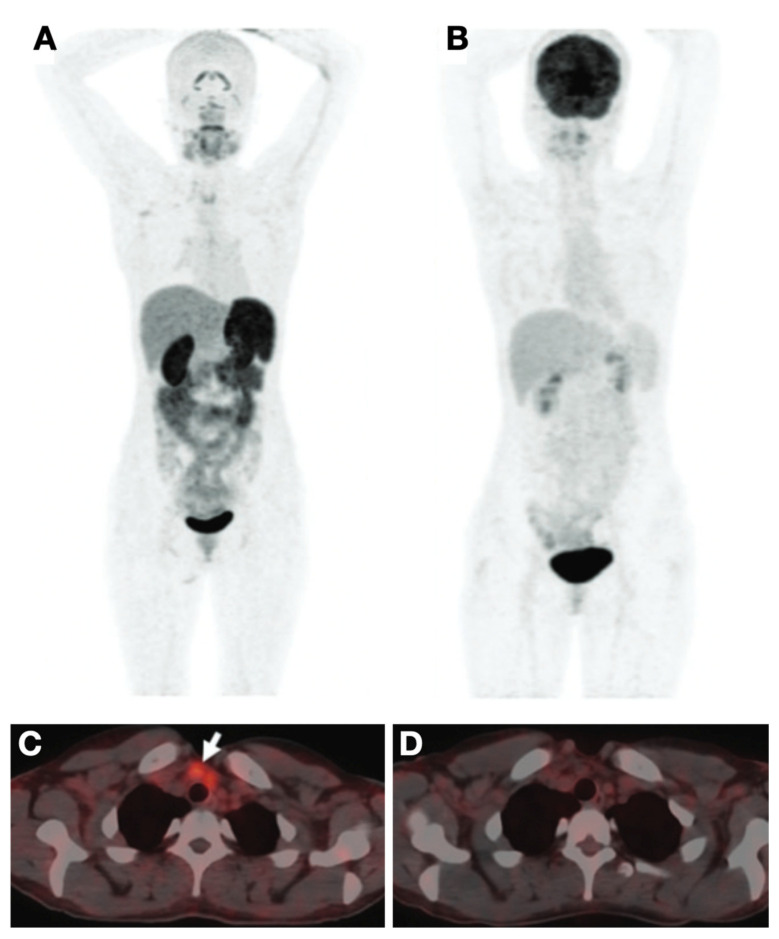
PET/CT imaging comparing [^68^Ga]Ga-DOTA-RGD2 and [^18^F]FDG. The RR-DTC case showed a high level of stimulated Tg (85 ng/mL) and negative ^131^I post-therapy whole-body scan. Histopathology confirmed the metastatic lesions. (**A**) A maximum intensity projection (MIP) image showed [^68^Ga]Ga-DOTA-RGD2 positive foci in the lower cervical region. (**B**) The corresponding MIP image of [^18^F]FDG. (**C**) Fused PET/CT showed the metastatic foci was [^68^Ga]Ga-DOTA-RGD2 positive; (**D**) the corresponding foci is negative on [^18^F]FDG fused PET/CT. Reproduced with permission from [85], copyright 2020 Mary Ann Liebert Inc.

**Figure 6 cancers-13-03188-f006:**
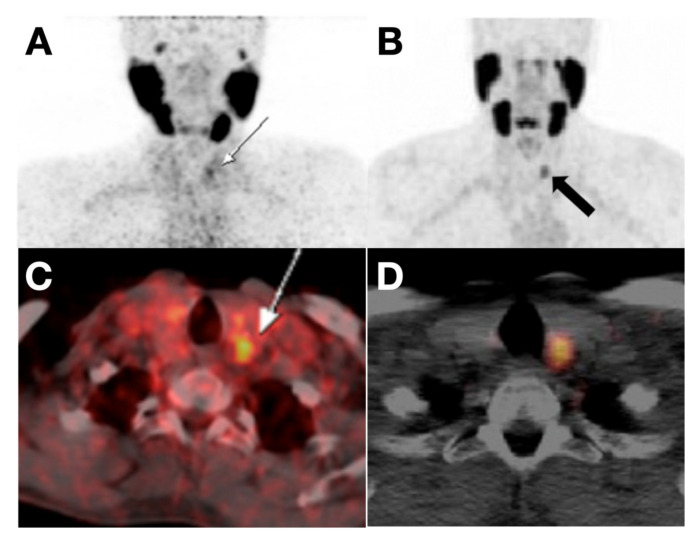
(**A**,**C**) PET and PET/CT fusion images showed slight [^68^Ga]Ga-PSMA uptake in the thyroid nodule of a 62-year-old patient with prostate cancer. The thyroid nodule was validated as Hürthle cell angioinvasive FTC by post-thyroidectomy pathology. Reproduced with permission from [88] copyright 2016 Society of Nuclear Medicine and Molecular Imaging. (**B**,**D**) PET and PET/CT fusion image marked ^68^Ga-PSMA uptake in the thyroid nodule of a 65-year-old man with metastatic prostate cancer. The thyroid nodule was regarded as TC proved by post-thyroidectomy pathology. Reproduced with permission from [91], copyright 2017 Wolters Kluwer Health Inc.

**Figure 7 cancers-13-03188-f007:**
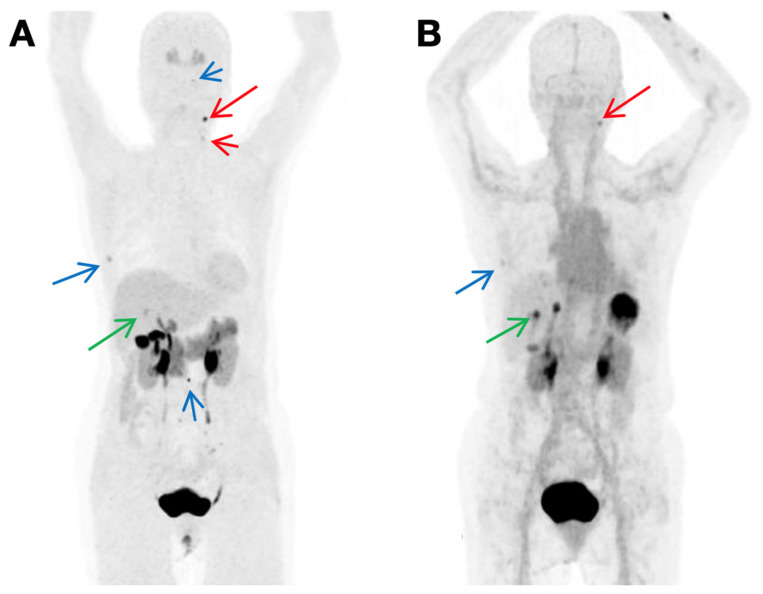
Comparison of [^18^F]FDOPA and [^68^Ga]Ga-DOTA-MGS5 in MTC patients. (**A**) [^18^F]FDOPA PET imaging at one hour post-injection. (**B**) ^68^Ga-Ga-DOTA-MGS5 PET imaging at one-hour post-injection. [^68^Ga]Ga-DOTA-MGS5 yields complementary information to [^18^F]FDOPA-PET. Reproduced with permission from [96], copyright 2021, Springer Nature Inc.

**Figure 8 cancers-13-03188-f008:**
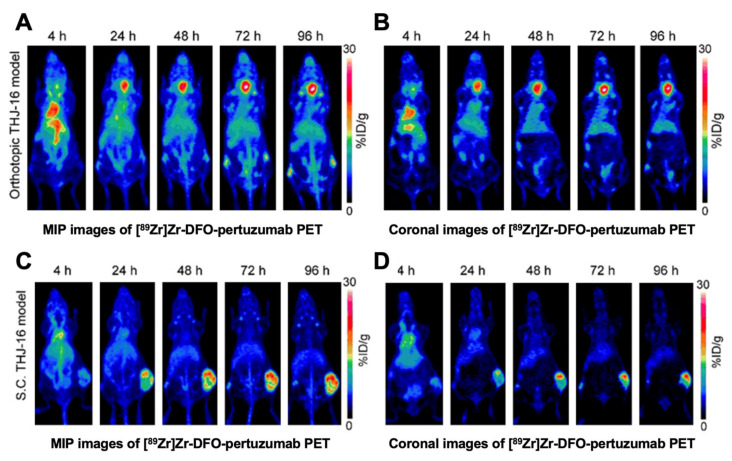
PET imaging with [^89^Zr]Zr-DFO-pertuzumab in xenografts (cell line: THJ-16T). (**A**) Maximum intensity projection (MIP) showed the ability of [^89^Zr]Zr-DFO-pertuzumab for visualizing TCs in an orthotopic model. (**B**) Coronal imaging in an orthotopic model. (**C**) MIP in a subcutaneous model. (**D**) Coronal imaging in a subcutaneous model. Reproduced with permission from [114], copyright 2019 e-Century Publishing Corporation.

**Figure 9 cancers-13-03188-f009:**
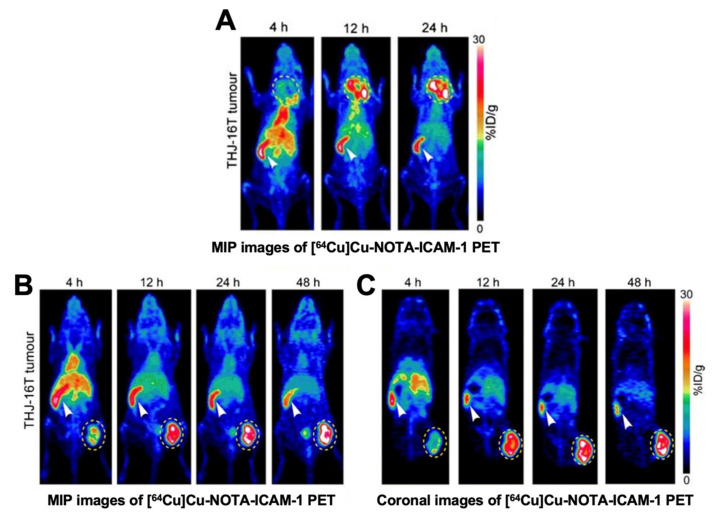
[^64^Cu]Cu-NOTA-ICAM-1 immunoPET imaging in ATC xenografts (cell line: THJ-16T). (**A**) Maximum intensity projection (MIP) images in an orthotopic model. (**B**) MIP images in a subcutaneous model. (**C**) Coronal images of the same model. Reproduced with permission from [101], copyright 2020 Springer Nature Inc.

**Figure 10 cancers-13-03188-f010:**
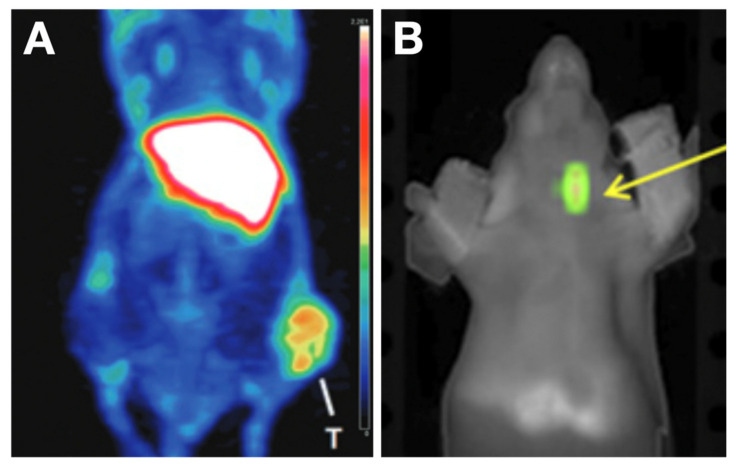
Characterization of [^89^Zr]Zr-Gal-3 in TC xenografts. (**A**) PET image acquired at 48 h post radiotracer injection in a subcutaneous TC model showed an apparent accumulation of [^89^Zr]Zr-Gal-3 in a tumor at the right thigh; Reproduced with permission from copyright 2016 American Association for Cancer Research [124]. (**B**) TC in an orthotopic model was visualized after injection of Cy5.5-Gal-3 with fluorescence imaging in the neck. Reproduced with permission from [125] copyright 2019 Society of Nuclear Medicine and Molecular Imaging.

**Figure 11 cancers-13-03188-f011:**
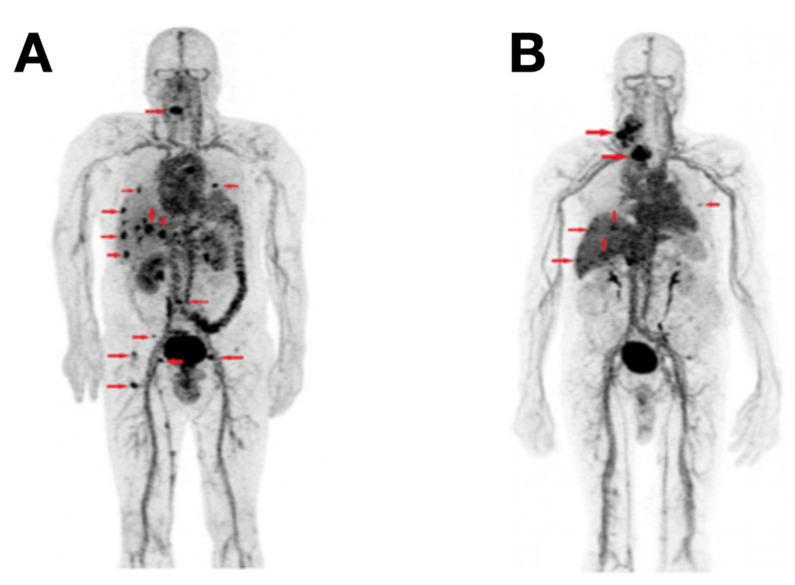
[^68^Ga]Ga-IMP288 plus TF2 PET revealed a considerable number of MTC foci. (**A**) Patient #1: foci were detected in multiple places, including supradiaphragmatic nodes, lung, liver, and bone, etc. (**B**) Patient #2: foci were detected in supradiaphragmatic nodes and liver, etc. Reproduced with permission from [134], copyright 2016 Society of Nuclear Medicine and Molecular Imaging.

**Figure 12 cancers-13-03188-f012:**
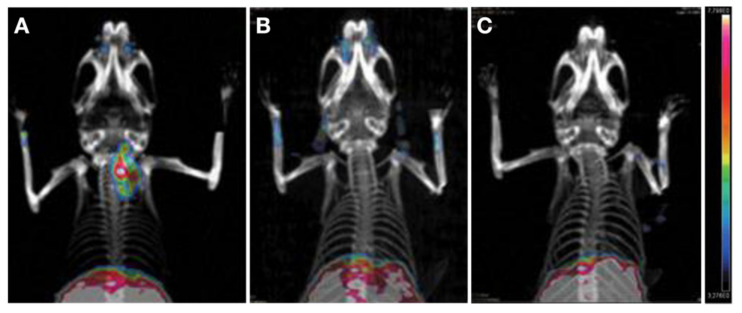
PET/CT images of mice at 24 h after intravenous injections. (**A**) Injection with 3 MBq of [^89^Zr]Zr-DFO-αGal3-Fab-PAS_200_. (**B**) Co-injection of 3 MBq of [^89^Zr]Zr-DFO-αGal3-Fab-PAS_200_ and 1000-fold of nonradioactive aGal3-Fab-PAS_200_ (for blocking). (**C**) Control. Color scale bars: 3.3–7.8%ID/g. Reproduced with permission from [139], copyright 2020 Mary Ann Liebert, Inc.

**Figure 13 cancers-13-03188-f013:**
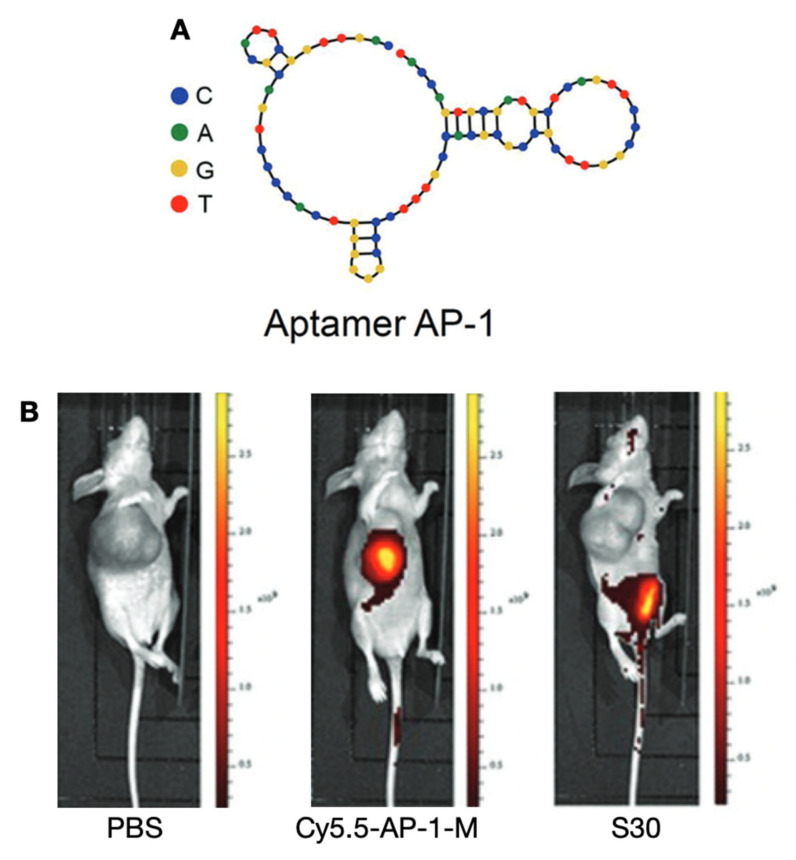
(**A**) Predicated structure of aptamer precursor AP-1. (**B**) Distribution of AP-1-M in a FRO xenograft model on fluorescence imaging at 48 h post-injection of PBS, Cy5.5-labeled AP-1-M, and control aptamer S30. Reproduced with permission from [15], copyright 2013 Royal Society of Chemistry.

**Figure 14 cancers-13-03188-f014:**
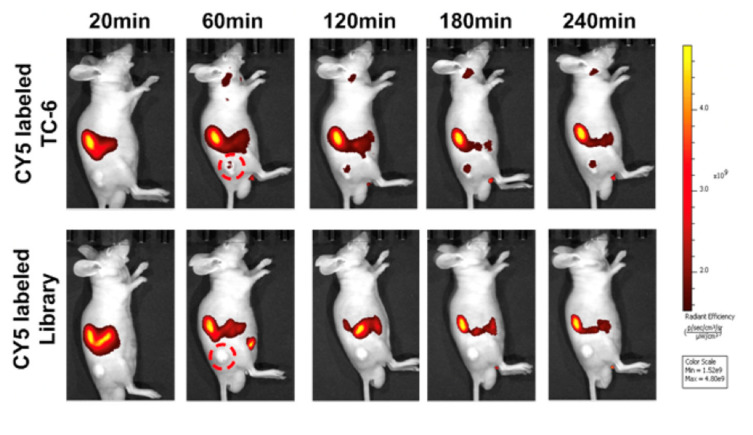
Time-lapse fluorescence imaged post-injection of Cy5-labeled TC-6 (**upper**) or library control (**lower**) in a TPC1 xenograft model. Reproduced with permission from [16], copyright 2016 American Association for Cancer Research.

**Figure 15 cancers-13-03188-f015:**
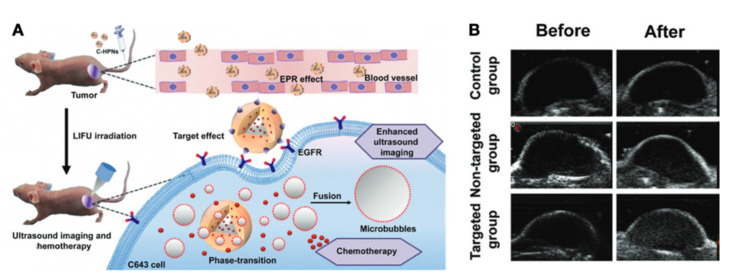
Theranostic applications of nanoparticles targeting EGFR. (**A**) Schematic illustration of the nanoparticles for chemotherapy drug delivery and enhanced diagnosis via LIFU. (**B**) Ultrasound imaging of tumors in B-mode before and after LIFU treatments. Reproduced with permission from [154], copyright 2019 Royal Society of Chemistry.

**Figure 16 cancers-13-03188-f016:**
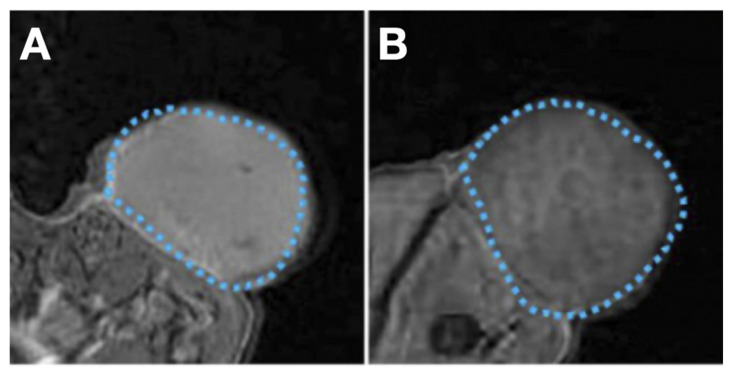
MRI in subcutaneous thyroid squamous cell cancer (TSCC) xenograft (blue dashed line) with NPs-SHP2 nanoparticles after LIFU treatment. (**A**) injection of SHP2-targeted nanoparticles. (**B**) injection of non-targeted nanoparticles (control). Reproduced with permission from [155], copyright Dove Medical Press Inc.

**Table 1 cancers-13-03188-t001:** A synoptic view of the MI radiotracers and their potential clinical value.

Tracer Types	Target	TC Type	Typical Roles	LoE [Ref.]
**Transporter-targeting probes**				
[^124^I]NaI, [^131^I]NaI	NIS	DTC	LL, TS, PE, TT, RD	Clinical [35,36]
[^18^F]TFB, [^18^F]FS, [^18^F]HFP	NIS	DTC	LL, TS, PE, IUE	Clinical [50,51]
[^18^F]FDG	GLUT1	TC	LL, TS, PE, PIU	Clinical [5,54,55]
[^18^F]FDOPA	SLC7A5, SLC7A8	MTC	LL, TS, PE	Clinical [59,65]
[^11^C]MET	SLC7A5	FTC	LL, TS, PE	Clinical [63]
[^18^F]FGln	SLC7A5, SLC1A5, SLC38A1	PTC	LL, TS, PE	Clinical [62]
[^18^F]FLT	ENT1	DTC	LL, TS, PE	Clinical [79,160]
**Peptide-based probes**				
^68^Ga/^177^Lu/^90^Y/^111^In labelled somatostatin analogue	SSTRs	TC	LL, TS, PE, PRRT	Clinical [161,162,163]
^68^Ga/^177^Lu labelled RGD_2_	αvβ3	TC	LL, TS, PE, PRRT	Clinical [85,86]
^68^Ga/^177^Lu labelled PSMA-ligand	PSMA	DTC	LL, TS, PE, PRRT	Clinical [88,91,164,165]
^68^Ga/^177^Lu labelled MGS5	CCK2R	MTC	LL, TS, PE, PRRT	Clinical [96] and preclinical [95]
**Antibody-based probes**				
Single target IgG-based probes				
^89^Zr or RDye 800CW labeled pertuzumab	HER2	ATC	LL, TS	Preclinical [114]
^64^Cu or RDye 800CW labeled ICAM-1 Ab	ICAM-1	ATC	LL, TS	Preclinical [101]
^89^Zr labeled Gal3 Ab	Gal3	DTC	LL, TS	Preclinical [124]
Bispecific IgG-based probes				
^68^Ga/^177^Lu labelled IMP288 plus TF2 BsAb	CEA × HSG	MTC	LL, TS, PE, PRRT	Clinical [134,166]
Fab-based probes				
^89^Zr or Cy5.5 labeled αGal3-Fab	Gal3	DTC	LL, TS	Preclinical [125,139,140]
Nanobody-based probes				
Targeting TROP-2 probes	TROP-2	TC (SP [167])	LL, TS, PRRT	Preclinical (OS)
Targeting CD47 probes	CD47	TC (SP [168])	LL, TS, PRRT	Preclinical (OS)
Targeting CD146 probes	CD146	TC (SP [169])	LL, TS, PRRT	Preclinical (OS)
**Other probes**				
Aptamer-based probes				
Cy5.5-AP-1-M	CD133	ATC	LL, TS	Preclinical [15]
Cy5-TC-6	PTC tissue	PTC	LL, TS	Preclinical [16]
Nanoparticles-based probes				
C-HPNs	EGFR	ATC	LL, TS, TT	Preclinical [154]
NPs-SHP2	SHP2	TC	LL, TS	Preclinical [155]

MI, molecular imaging; SP, speculative; LoE, level of evidence; LL, lesion localization; PE, prognosis evaluation; TT, tumor therapy; RD, radioiodine-131 dosimetry, [^131^I]NaI dosimetry; TS, tumor staging; IUE, iodine uptake evaluation; PIU, predicting iodine uptake; OS, ongoing study; PRRT, pre-evaluation for receptor radionuclide therapy; TC, thyroid cancer; PTC, papillary thyroid cancer; MTC, medullary thyroid cancer; ATC, anaplastic thyroid cancer; DTC, differentiated thyroid cancer; BsAb, bispecific antibody; Ab, antibody; Fab, antigen-binding fragment.

## Data Availability

No new data were created or analyzed in this study. Data sharing is not applicable to this article.

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
