# Peer review of "Next-Generation Molecular Imaging of Thyroid Cancer"

_cancers, 2021, doi:10.3390/cancers13133188_

Round 1

Reviewer 1 Report

  • Authors are encouraged to actualize their bibliography with the latest publications from 2021 on this topic
  • P.2 l.84 (and subsequent uses of "-based" in this paragraph for radiotracers that use a transporter to accumulate in cells) : Replace "2. Transporter-based probes" by "2. Transporter-targeting probes". Using "based" underlines that the structure of the probe is based on the transporter (instead of its ligand).
  • p.7 l.241: replace subparagraph title by "Somatostatin receptor (SSTR)-targeting probes"
  • p.7 l.251: replace subparagraph title by "avb3 integrin-targeting probes"
  • p.15 l. 470 : Not sure it is worth mentioning this devlopment given the tumor size on the figure...
  • p.16 l. 487 : "Nanoparticles-based probes"
  • Radiotracer nomenclature should follow recommendations from EJNMMI
  • Many abbreviations are not explicited the first time they are used
  • Please summarize each section in a table, aiming at a synoptic view of the TC type, diagnostic and/or prognostic value of the radiotracer, level of evidence (preclinical / clinical)...
  • Reference list : the reference number figures twice

Reviewer 2 Report

In this article, Jin et al. review the recent literature (approx. 2012-2021) on advances in molecular imaging related to thyroid cancer. They provide a review organised around the chemical nature of molecular imaging probes (enzyme/transporter substrates, peptides, antibodies and others, e.g. nanoparticles and aptamers), further divided into various targets. Overall this is an informative review, relatively well structured and written (although some syntax/grammatical errors need to be corrected, see below in minor comments). It should be acceptable for publication with a few revisions, detailed below.

Major comments:

  1. L41-83: The introduction could benefit from a few more details regarding the diagnosis and treatment options for TC. 131I treatment and surgery are only briefly mentioned. Reviews have an educational role, so explaining these aspects in a few lines and pointing to more detailed reviews would benefit the manuscript. The authors have longstanding expertise in molecular imaging, but most readers of this journal may not be so well-versed in MI. It could be helpful to add a reference or two to some introductory literature (the “Molecular imaging primer” by James&Gambhir, 2012, for example).

  2. L95-107: Similar issue, I think this assumes too much knowledge from readers who do not have a background in nuclear medicine or imaging. There are several medically useful radioisotopes of iodine but this section just lists two without explanation. What differentiates them and what are they used for? A few sentences would help. This paragraph should also explain why there is a need to increase NIS expression (i.e. because its expression is downregulated in radiation-resistant TC), and why this transporter in particular (owing to its physiological role in the thyroid and limited expression in the rest of the body, whereas most of the other targets mentioned in the review are usually present at low levels in many tissues and overexpressed in many types of cancers).

  3. L127-140: Can the authors further explain what they mean by “TFB is an analog to radioiodine yet different”? This is too vague. It should mention for example that TFB and HFP have the same charge and similar ionic radius to iodide and therefore can be transported by NIS. Furthermore, since they enter thyroid cells by the same mechanism as iodide, how would such tracers be helpful in TENIS? This is not intuitive. Dittmann2020 provide an interesting discussion of this issue and a few of these arguments could figure in this manuscript. Overall this paragraph feels a bit disorganised and could be reworked. For completeness it should also mention [18F]SO3F- (Khoshnevisan et al, J Nucl Med 2017).

  4. L260 and 275-276: There is a mention of 177Lu and 225Ac (for PRRT although this is not explicitly written). In general, the review in its current form is missing a good opportunity to introduce and discuss the concept of theranostics (and radioisotope pairs using the same probe and chelator for imaging and therapy) and its utility in thyroid cancer. This could be added in the form of a separate paragraph and would greatly benefit the reader. There is also a body of work around 188Re that is not mentioned at all.

  5. L291-292: the authors mention “impressively high uptake” of 68Ga-DOTA-MGS5. I am not a fan of emphatic language but more importantly, the original data and comments from the original authors are far more nuanced. I suggest rewording for balance.

  6. L303: This sentence is unclear: how does the “unmatched ability to identify targets” hinder the use of antibodies? (I presume this refers to the high affinity of monoclonal antibodies for their targets, but I still don’t get the point).  Please reformulate and clarify.

  7. In section 4 (antibody probes): many (most?) antibody-based imaging probes have simply been prepared by non-selective conjugation on lysine/cysteine residues with chelators (or other radioactive/fluorescent tags). It is well known that this approach can affect target binding affinity, especially at high conjugate/protein ratio, and in any case leads to a mixture of products with different numbers of tags per protein molecule. Considering all the advances that have been made in protein engineering, I think it would be good to advocate (to the people designing antibodies, nanobodies etc) more strongly for the systematic inclusion of tags that enable site-specific conjugation. This would facilitate the development of (better) imaging probes and especially companion diagnostic agents for biological drugs. The authors mention this in the conclusion (L551-552) but I think they could be even more specific. In the current situation, imaging seems to be treated as a second thought (“oh well, I guess we could always radiolabel it”). I leave this point to the discretion of the authors.

  8. General comments for discussion: in the last 2 decades there has been a general increase in prevalence of TC. This is partially due to environmental factors, but also due to overdiagnosis (increased screening with more sensitive methods, especially in developed countries), leading to unnecessary treatment. How can molecular imaging avoid this pitfall, what are the risks that developing even more sensitive molecular imaging techniques will lead to overdiagnosis and overtreatment?

 Minor comments and suggestions:

  • L42: “most common types of cancer”: please state a percentage value or range (e.g. TC represents x% of new cancer cases)
  • L49: “Within” is more correctly followed by “of”, ie “Within ten years of an initial […]”
  • L70: what do the authors mean by “a proteome signature”?
  • L77-83: this paragraph would be better written in the present tense
  • L88-89: suggested rephrasing: “Several alternatives to radioiodine have been discovered to identify RR-DTC metastases lacking radioiodine uptake”
  • L90: please indicate the chemical form of 201Tl for TC imaging (as it is indicated for the other radiotracers)
  • L121: “implicating” is not the right word here. Suggested replacement: “Another ERRγ inverse agonist, DN200434, was recently shown to increase radioiodine uptake in ATC tumors, identifying ERRγ as a target […]”
  • L127: error in section title formatting
  • L138: ref 33 is incorrectly used as it refers to HFP, not TFB
  • L181-182: here add “LAT1” abbreviation as it is used further in the text
  • L183 and also L219: “As of yet” is not a great expression. Suggested replacement: “to date”.
  • L191: suggested corrections: “[…] in clinical settings [54]. The downside of 11C-MET is the short half-life of 11C (20.4 min), which complicates its use.”
  • L216: since there is only one example, there is no need for a subsection
  • L223: suggested correction: “69% of metastatic lesions were identified by focal increases in 18F-FLT uptake, a lower result than the 92% identified by 18F-FDG PET/CT.” Furthermore, the mention of FLT in immunotherapy feels a bit out of scope here because the authors don’t comment on immunotherapy elsewhere in the manuscript.
  • L242: suggested rephrasing and additional reference: “Somatostatin receptors, which are therapeutic targets in neuroendocrine tumors (NETs), are often overexpressed on the surface of tumour cells. This has led to the development of several 68Ga-labelled somatostatin analogs as PET imaging probes (Pauwels et al, Am J Nucl Med Mol Imag 2018), which could be used for diagnosis MTC”.
  • L258-259: This sentence is unclear: which tracer is incremental over the other? Suggested rephrasing: “the advantage provided by 68Ga-DOTA-RGD2 is the ability to detect lesions not detected by 18F-FDG” or similar (if I understood correctly).
  • L316-319: suggested rephrasing: “[…] multiple antibodies and antibody derivatives have been designed as imaging probes and as therapeutic agents to induce cancer cell death and elicit host immune effector responses in TC [80].”
  • L334: This sentence is out of place because the previous ones already mention TC, but not breast cancers. Please re-order or re-phrase.
  • L376: Can the authors elaborate of what makes 89Zr-Gal-3 “an excellent candidate”? It appears this antibody has been carefully designed so perhaps some further details are warranted.
  • L386: since there is only one example, there is no need for a subsection.
  • L423: since there is only one example, there is no need for a subsection
  • L432-436: in Figure 12 caption, harmonise spelling of DFO (all caps as in the rest of the manuscript)
  • L463: typo error in section 5.1.1 title (CD133)
  • L494: this sentence is a bit vague, it suggests there is something specific to TC that makes antibodies/peptides good choices for nanoparticle targeting (whereas I think this is the case for most cancers). I would delete that sentence and replace the following one by “Although applications of targeted nanoparticles in TC have so far been limited, there have been publications investigating nanoparticles conjugated to antibodies targeting EGFR or SHP2 [132,135]”.
  • L507-508: suggested correction: “The core explosion induced by PFP boiling causes the release of 10-HCPT, providing a more targeted delivery of the chemotherapeutic drug [135]”.
  • L529: the advantages of MI in diagnosis are clear but has it actually led to changes in what treatments the patients are given?
  • L530: suggested correction: “immune-based probes generally have high specificity”
  • References 43 and 138 are quite old, please consider replacing with more recent literature.

Round 2

Reviewer 1 Report

Thanks for taking our comments into account.

Author Response

Thank you for your input.

Reviewer 2 Report

(see comments in Word document)
